



# Aerial and surface rivers: downwind impacts on water availability from land use changes in Amazonia

Wei Weng[1,2,3], Matthias K. B. Luedeke[1], Delphine C. Zemp[1,4], Tobia Lakes[2,3], Juergen P. Kropp[1,5]

[1]Potsdam Institute for Climate Impact Research, Potsdam, 14482, Germany
[2]Geography Department, Humboldt-Universität zu Berlin, Berlin, 10099, Germany
[3]Integrative Research Institute on Transformations of Human-Environment Systems, Humboldt-Universität zu Berlin, Berlin, 10099, Germany
[4]Biodiversity, Macroecology & Conservation Biogeography, University of Goettingen, Goettingen, 37077, Germany
[5]Institute of Earth and Environmental Science, University of Potsdam, Potsdam, 14469, Germany

*Correspondence to*: Wei Weng (weiweng@pik-potsdam.de)

**Abstract.**

The abundant evapotranspiration provided by the Amazon forests is an important component of the hydrological cycle both regionally and globally. Since the last century, deforestation and expanding agricultural activities have changed the ecosystem and its provision of moisture to the atmosphere. However, it remains uncertain how the ongoing land use change
will influence the rainfall, runoff, and water availability as findings from previous studies differ. Using moisture tracking experiments based on observational data, we provide a spatially detailed analysis recognising potential teleconnection between source and sink regions of atmospheric moisture. We apply land use scenarios in upwind moisture sources and quantify the corresponding rainfall and runoff changes in downwind moisture sinks. We find spatially varying responses of water regimes to land use changes which may explain the diverse results from previous studies. Parts of the Peruvian
Amazon and western Bolivia are identified as those sink areas most sensitive to land use change in the Amazon and we highlight the current water stress by Amazonian land use change on these areas in the water availability. Furthermore, we also identify the influential source areas where land use change may considerably reduce a given target sink's water reception (from our example of the Ucayali river basin outlet, rainfall by 5–12 % and runoff by 19–50 % according to scenarios). Sensitive sinks and influential sources are therefore suggested as hotspots for achieving sustainable land–water
management.

## 1 Introduction

The Amazon basin, draining about 7 million km$^2$, is the largest river basin in the world. It hosts the most extensive tropical rainforests ecosystem covering about 5.3 million km$^2$, which represents 40 % of the global tropical forest area (Laurance et al, 2001; Aragão et al., 2014). The substantial transpiration from the canopy in addition to the evaporation
contributes to abundant water fluxes to the atmosphere (Fisher et al., 2009). This atmospheric moisture eventually returns to the land and contributes about 25–35 % of the basin's and 48–54 % of the region's rainfall (Salati and Nobre, 1991; Eltahir



and Bras, 1994; Trenberth, 1999; Bosilovich and Chern, 2006; van der Ent et al., 2010, Zemp et al., 2014). Regulating the water cycle in the region, the Amazon forests are a key component of both the regional and global climate system (Foley et al., 2003, 2005; Meir et al., 2006; Snyder et al., 2010; Anderson-Teixeira et al., 2012).

This ecosystem and its climate regulations operate under uncertainties of the undergoing land use change (Pielke et al., 2002, Foley et al., 2007; Chapin et al., 2008; Soares-Filho et al., 2014). Since the 1960's, there has been substantial clearing of the Amazon forest for agricultural purposes, about 15 % of Brazilian Amazon rainforests have been cleared (INPE, 2017). Deforested areas are most often used as pasturelands, e.g. 80% of the cleared areas are converted into pasturelands (Veiga et al., 2002). Rice, cassava, and, to a lesser extent, maize and soybean have also driven deforestation (Nepstad et al., 2006; Barona et al, 2010). Soarse-Filho et al. (2006) have projected a loss of 47 % Brazilian rain forest cover by 2050 under a business as usual scenario compared to the situation in 2004. Although this fast Brazilian deforestation trend has decelerated since 2004, a rebound of the deforestation rate has been observed since 2013 (Hansen et al., 2013; INPE, 2017). Moreover, a more recent Brazilian forest policy shift may allow for further deforestation in the country (Soares-Filho et al., 2014; Aguiar et al., 2016) in addition to observed increases in deforestation rates in other Amazonian countries (Hansen et al., 2013).

Through land–atmosphere coupling mechanisms, deforestation and other land use changes in the Amazon affects climate both regionally and globally (Dickinson and Henderson-Sellers, 1988; Dirmeyer and Shukla, 1994; Gedney and Valdes, 2000; Costa and Foley, 2000; Snyder, 2010). Among those changes, modified moisture fluxes to the atmosphere (Gordon et al., 2005) introduce shifts in rainfall pattern and runoff regime, and influence water availability (Henderson-Sellers et al., 1993; D'Almeida et al. 2007; Coe et al., 2011; Bagley et al., 2014; Lima et al., 2014; Swann et al., 2015; Spracklen and Garcia-Carreras, 2015). Given the spatial differences found in land–atmosphere coupling strength (Koster et al., 2004; Seneviratne et al., 2006) and continental moisture recycling (van der Ent et al., 2010), some areas' water regime are more sensitive to land use change than others. However, this spatially different sensitivity in the hydrological responses to land use change is not well-understood. Indeed, the water regime changes are also experienced by the downwind regions that are spatially displaced from where the land use change is taking place. Thus, it requires investigation into both the sinks and the sources of the moisture flows to understand this spatial difference, which has not been covered in depth by previous studies. Such an investigation will advance the understanding of land use change impacts on the water cycle and is necessary in order to identify hotspots for conservation policy targets fulfilling the SDGs.

The most direct way of portraying the airborne moisture flows is using diagnostic models driven by observation data (or observation-based climatic data for data scarce regions). In the present study, we utilise a moisture recycling tracking algorithm to structure the moisture flow for exploring spatial heterogeneity in land use change impacts on the rainfall and runoff in Amazonia. Moisture recycling describes the contribution of local evaporation to local precipitation and was investigated in earlier studies by utilising bulk models to partition moisture recycling in the water cycle within an area of interest (Brubaker et al., 1993; Eltahir and Bras, 1996; Trenberth, 1999). Moisture tracking tools have been further developed to describe the course in which evapotranspirated moisture travels through the atmosphere and precipitates in downwind regions within the area of interest, thus making perceivable the architecture of aerial rivers (termed in Arraut et al.,





(2012) as preferential pathway of moisture flow, a good analogy with the surface river). Moisture tracking recognises tele-connections between moisture sources and sinks, which are not limited to administrative and topographical boundaries. These moisture tracking tools include isotopic tracers (Salati et al., 1979; Victoria et al., 1991; Henderson-Sellers et al., 2002; Tian et al, 2007), numerical algorithms online coupled with an atmospheric circulation model (Koster et al, 1986; Bosilovich and Chern , 2006), or offline a posteriori with reanalysis or operational data (Yoshimura et al., 2004; Dirmeyer and Brubaker, 2007; van der Ent et al., 2010; Tuinenburg et al., 2012; Spracklen et al., 2012; Bagley et al., 2014). Here we use an off-line Eulerian numerical tracking algorithm (WAM-2layers, van der Ent et al., 2014, see also Sect. 2.1.1) driven by observation based data to approach moisture flows. While having relatively low computation cost, it is robust in identifying the spatial pattern of moisture flow in a certain region (Keys et al., 2012) and the regional moisture recycling simulation of the Amazon basin by this method is in line with other moisture tracking approaches (Table 2; Zemp et al., 2014).

Our objectives are (1) to explore how land use change impacts on rainfall and runoff in Amazonia can differ spatially, (2) to quantify this spatial variation and (3) to identify the sensitive regions to Amazonian land use change.

To address these objectives, spatially different rainfall and runoff responses at moisture sinks are quantified when land use change occurs in Amazonia. Different hydrological influences that result from land use change in various moisture source areas are also calculated. Furthermore, we identify the sensitive sinks (land surface areas where the water regime is most impacted by land use change in a given upwind area via moisture recycling) and the influential sources (land surface areas where land use change exerts the strongest impacts through moisture recycling on the water regime of a given area downwind).

In the following section we describe the moisture tracking experiments and the scenarios that were utilised to analyse land use change impacts on water regime. We also introduce the concept of the most influential precipitationsheds (MIPs), which is used for highlighting the influential sources of moisture. In Sect. 3, we present the results of identification of sensitive pairs of sinks and sources to Amazonian land use change. Then, we present the quantified impacts on rainfall and runoff by land use change in terms of sensitive sinks and influential sources. Additionally, calculation of upper bound water regime changes from hypothetical whole-Amazon land use changes are also shown for further comparison. We discuss implications of our results in Sect. 4. These include the contribution of the interconnection between surface and aerial rivers to the spatial heterogeneity and the importance of aerial river conservation hotspots when compared with the upper bound. We highlight the current pressure on the sensitive regions' water availability by land use change. The uncertainties and limitations of our results are also discussed in this section. In Sect. 5, we conclude our findings and show how they resonate with the current discussion in the field. We then provide suggestions on managing land use change impacts on the water availability for sustainable land–water use in Amazonia.



## 2 Methods

### 2.1 Outlining aerial rivers

#### 2.1.1 Tracing moisture flow in Amazonia

The moisture flow is traced by the Water Accounting Model-two layers, WAM-2layers version 2.3.01 (van der Ent et al.,
2014) for the South American continent. With an Eulerian specification of the field, the WAM-2layers model backtracks the
moisture origin of precipitation that occurs over a given area following water balance. The backtracking is based on given
input data while assuming that the water reservoirs of the lower atmospheric layer and the land surface are well mixed.

The WAM-2layers distinguishes the bottom and top atmospheric layers (approximately 800hPa for a standard surface
pressure) in the calculation of moisture flux across grid cell boundaries (van der Ent et al., 2014). This allows for the better
capture of the wind shear system that resulted in errors in traditional offline 2-D tracking studies with a well-mixed
atmosphere assumption (Goessling and Reick, 2013; van der Ent et al., 2013).

We use simulations from WAM-2layers from a previous moisture back-track modelling experiment (MOD experiment,
see Zemp et al. (2014)). The WAM-2layers model run for the MOD experiment was on a 1.5° latitude–longitude grid and the
time coverage was 2000–2010. The input data of the first year was used for spin-up runs. The MOD experiment result
further used in this study is the moisture transport matrix $\boldsymbol{m}$. Its elements $m_{ij}$ describe the amount of moisture
evapotranspired from grid cell $i$ which is precipitated in grid cell $j$.

#### 2.1.2 Input data

The input data for evapotranspiration (E) and precipitation (P) of the MOD experiments is based on global satellite
products (see Table 1). The evapotranspiration input was derived from the Moderate Resolution Imaging Spectroradiometer
(MODIS) evapotranspiration product MOD16ET (Mu et al., 2013). Based on the Penman–Monteith equation and the
algorithm from Cleugh et al. (2007), global evapotranspiration is calculated as the sum of evaporation (from different soil
types and interception by the canopy) and transpiration from the vegetation while environmental constraints and diurnal
cycles are recognised. The calculation is based on MODIS Earth observation data inputs (land cover, albedo and enhanced
vegetation index) in conjunction with the Global Modeling and Assimilation Office (GMAO, v.4.0.0) daily meteorology data.
Loarie et al. (2011) validated MOD16ET's estimation with eddy flux tower data and reported its good performance
(differences in annual average of evapotranspiration are less than 4 % in savannas, 5 % in tropical forests and 13 % in
pasture agricultural lands). The precipitation input used in the MOD experiment was the product from the Tropical Rainfall
Measuring Mission (TRMM) Multisatellite Precipitation Analysis (TMPA) algorithm 3B42 version7, in which rainfall data
is acquired from multiple satellite observations including passive microwave and infrared data, which were then calibrated
by global rain gauge data (Huffmann et al., 2007). This remote sensing based rainfall data is widely used in regions that lack
ground observations such as the Amazon (Wagner et al., 2009; Su et al., 2008; Awadallah and Awadallah, 2013). This
dataset has been described as robust in precipitation estimations over the Amazon region especially at a monthly time scale



(Su et al., 2008; Collischonn et al., 2008). Humidity and wind speeds were taken from the ERA-Interim reanalysis product (Dee et al., 2011). Input data has been upscaled to the spatial resolution of the WAM-2layers model and downscaled to a temporal resolution of 3 h using the temporal variability in the corresponding ERA-interim products.

### 2.1.3 Structuring the Precipitationsheds: the MIPs

In our analysis, we utilize the concept of precipitationsheds and outline them for our target areas according to $m_{ij}$, the amount of moisture evapotranspirated from grid cell $i$ which is precipitated in grid cell $j$, derived from the MOD experiment as described in Sect. 2.1.1. The concept of precipitationshed was introduced by Keys et al. (2012) as the upwind surface areas providing evapotranspiration to a specific area's precipitation. In the present study we focus on the terrestrial component of precipitationsheds because of their relevance to land use change. Inter-continental moisture transports are

neglected as they have little influence in our study region (van der Ent et al., 2010). Recognising the spatial heterogeneity of the contribution in the precipitationshed, we further extract the Most Influential Precipitationsheds (MIPs) for our analysis. The MIP includes the most prominent contributing source areas of the evapotranspiration and therefore governs a given proportion of the sink area's precipitation with minimum land surface areas. An example of MIP for a grid element located in the Yurimaguas area is depicted in Fig. 1. The areas delimited by the 0.2 contour line is the smallest land surface

contributing to 20 % of precipitation in the Yurimaguas grid element from continental evapotranspiration. Out of this area, a wider land surface area collectively contributes to the same amount, the area between 0.2 and 0.4 contour lines or the area between 0.4 and 0.6 contour lines, for example. The area governs 20 % of continental moisture and is defined here as the 20 % threshold MIP for the Yurimaguas grid element. Likewise, the 40 % threshold MIP and the 60% threshold MIP are the areas delimited by the 0.4 contour line and the 0.6 contour line in Fig. 1. The larger the threshold value, the more

insignificant contributing source areas are included.

Under the modelling resolution of the present study, a 40 % threshold is the minimum contour value to delimit precipitationshed areas for some regions (eg. the Andes regions). Aiming to approximate the most influential precipitationshed by a standard that can apply to all the grid elements, the smallest valid 40 % threshold has been applied throughout our analysis for the identification of the MIPs.

### 2.2 Modified downwind precipitation by land use change

We employed different land use scenarios to investigate evapotranspiration shifts introduced by various land activities and their impacts on rainfall and runoff. The proportion of precipitation changes for the grid cell $j$ in a land use scenario that occurs in the region $\Omega_1$ can be described as

$$\Delta P_j = \sum_{i \in \Omega_1} m_{ij}\left(1 - \frac{E'}{E_i}\right) \tag{1}$$



Where $\Delta P_j$ stands for the changes in precipitation in sink grid cell $j$, $E_i$ is the original evapotranspiration in source grid cell $i$ which is located in the domain $\Omega_1$ and $E'$ is the corresponding evaporation of different land use types. This description is a first order approximation implying that major wind patterns remain similar when land use change occurs and feedback mechanisms such as altered energy balance, surface roughness and aerosols (Bonan et al., 2008; Mahmood et al., 2014) have

not yet been triggered or are of minor importance (Bagley et al., 2014). Empirical evaporation measurements of different land uses in the Amazon were derived from the Large-Scale Biosphere-Atmosphere Experiment (LBA-ECO) flux tower data (see Table 2) (Sakai et al., 2004). The LBA-ECO flux tower observation was established in 2000 in the Santarém region in the Brazilian Amazon. The field has been converted into different land uses including old-growth forest, selective logging, bare soil, pasture land, and rice cropping during the flux tower's operation period. The evaporation was estimated by the

Eddy Covariance (EC) method, corrected by the nocturnal boundary layer budget method for night time respiration underestimates, and validated by Acevedo et al. (2004).

Changes in the annual surface runoff regime by altered moisture recycling under land use change are investigated as well. By assuming that E and P are in balance (see Zemp et al. 2014) and steady groundwater storages, we use precipitation minus evaporation (P–E) to estimate annual surface runoff. We calculate the control state of P–E throughout catchments

using the 10 year average of the respective input data from the MOD experiment (2000–2010). The P–E changes under different land use scenarios are obtained by calculating the altered precipitation in the catchment grid cells and subtracting altered evaporation (E´) according to each land use scenario. The P–E values under different land use scenarios are then compared with the control state.

## 2.3 Sensitive pairs of sink and source regions

High precipitation sensitivity of a sink region regarding land use changes in its source regions combines two aspects: firstly, the precipitation in the sink region must depend strongly on aerial moisture transport from terrestrial sources (ie. high dependency on the aerial rivers) and secondly, the areal extent of the relevant source regions has to be rather small. The latter results in strong effects with even spatially limited land use changes. Given the importance of the Amazonian provision of moisture on the regional climate, we first calculate the precipitation recycled from the basin for each continental grid element.

In the following, we identify the grid elements with the highest ratios (defined by the 98 % percentile) of precipitation contributed by the moisture from the Amazon basin as sensitive sink areas. Next we determine the MIP (see Sect. 2.1.3) for the sensitive sink areas to examine their precipitation sensitivity to Amazonian land use changes.



## 3 Results

### 3.1 Sensitive sinks and influential sources: water regime shifts by upwind land use change

The most sensitive sinks regarding to the evapotranspiration of the Amazon basin are situated in the eastern foothills of the Andes, a geographical region in southern Peru and western Bolivia where over 70 % of the precipitation originates from
the Amazon, according to our results. The sensitivity to potential Amazonian land use change is shown in Fig. 2. The sensitivity increases westward throughout the Amazon forest and reaches its peak at its south-western fringe. We identified regions that have more than 50 % of the rainfall coming from Amazonian evapotranspiration (98 % percentile of the highest sensitivity to Amazonian land use change, called hereafter "sensitive areas"), and tracked back the location of the most influential sources for them as the second step in the procedure described in Sect. 2.3. It turns out that the south-western part
of the Amazon forest exerts the strongest influence. As demonstrated by Fig. 3, the most influential precipitationshed (MIP; the area delimited by the first contour line in Fig. 3) of the sensitive areas is located in the region Ucayali, Peru. This particular part of the Amazon forest governs the rainfall of the sensitive areas with high spatial efficiency compared to the rest of the moisture sources. While covering 3.5 % of the Amazonia, the MIP accounts for 50 % of the Amazonian evapotranspiration contribution to the sensitive areas' rainfall.

The above result on the most sensitive source and sink regions leads to the choice of interesting areas to quantify the influence of defined land use scenarios on precipitation and runoff regimes. As we are interested in the relationship of land use effects on both precipitation and surface runoff availability, we investigate them at the outlet of the Ucayali River basin (referred to as the target sink hereafter), a sub-basin where half of the sensitive areas are located (see Fig. 3). Accordingly, we applied land use scenarios in different spatial domains including the Ucayali River basin (the watershed of the target sink)
and the MIP of the target sink. In addition, land use scenarios are also employed to the MIP of the Ucayali river basin (the MIP of the watershed) but excluding the basin component cells in order to understand land use change influences outside of the watershed boundary, which is traditionally not covered in depth in water availability studies. Figure. 4 shows the location of different land use scenario domains.

Different land use scenarios including the conversion of the areas to bare soil, dry and wet pasturelands, and rice
cropping are applied in each domain depicted in Fig. 4. For each domain and each scenario, we investigate changes in the rainfall and runoff reception of the Ucayali River outlet, the target sink, as described in Sect. 2.2. Figure. 5 shows the interactions which are considered: Changes in evapotranspiration when applying land use scenarios influence the rainfall downwind in both the target sink (here the Ucayali River basin outlet) and the target sink's upstream watershed (here the Ucayali basin) through moisture recycling (the light blue arrows in Fig. 5), thus altering the rainfall and runoff reception in
the target sink. We note that the runoff changes measured in the target sink are also influenced by the land use scenario applied in the domain of the target sink's upstream watershed (here the Ucayali basin) as shown by the black arrows in Fig. 5.

Changes in the rainfall and runoff reception of the target sink vary in direction and magnitude when land use change occurs in different spatial domains (Table 3). Bare soil land use scenario leads to more considerable alteration than the





pasturelands and rice cropland scenarios, which have the least impact. Generally, the rainfall decreases when land use changes, but the extent depends on the location of such change. Land use change in the MIP of the target sink leads to a reduction on the target sink's rainfall by 5 % (rice cropping) to 12 % (bare soil). On the other hand, when land use change occurs in the Ucayali river basin, the rainfall in the target sink experiences a mild reduction of less than 5 % in all scenarios.

Runoff shifts differ in sign when land use change occurs in different locations. Increase in runoff received by the target sink is found when applying land use scenarios in the Ucayali basin: the runoff is intensified from adding a quarter (27 %, rice cropping) to adding an extra time (103 %, bare soil) on the original flow. However, we found that applying land use scenarios out of the watershed boundary has negative influences on the runoff of the target sink. Land use change in the MIP of the watershed results in 19 % (rice cropping) to 50 % (bare soil) reduction in target sink's runoff. The heterogeneous

hydrological response due to the location of land use change is discussed in Sect. 4.

**3.2 Upper bounds for the influences of Amazonian land use change**

So far we investigated the most sensitive source-sink pairs and chose the considered land use change areas accordingly. However, land use change may occur in various parts of the Amazon basin. Therefore, we estimated rainfall and runoff changes considering Amazon wide land use change to describe the upper bounds of land use change impacts on water

availability. For that, we apply in the following different hypothetical homogeneous land uses over the whole Amazon basin and calculate their effects on precipitation and runoff at different locations.

Table 4 shows the results for the reduction of rainfall in different Amazonian land use scenarios. Sensitive areas can experience 11.3–38.5 % annual rainfall reduction via moisture recycling when all of the Amazon forest is cultivated. The reduction in sensitive areas almost doubles the reduction of rainfall in the Amazon basin average (6.5-18.2 %) and it also

greatly surpasses the average southern American continent decrease in rainfall (4.0–12.9 %). The bare soil land use scenario results in the greatest reduction in rainfall while the rice cropping scenario exerts the least influence on rain reception in the sensitive areas. The same fashion appears in the continental and the Amazon basin average.

Conversely, runoff estimates rise in all land use scenarios but in different extent across sub-basins. As shown in Table 5, the bare soil land use scenario introduces the greatest increase (by 32.7 %) among all scenarios in the runoff of the Ucayali

river basin, a sub-basin where half of the sensitive areas are located. Rice cropping has a milder impact resulting in nearly a 1 % increase in the Ucayali runoff. The extent of the runoff increase is different across the basins. Runoff estimates of the Madeira basin, the largest sub-basin in the Amazon (see Fig. 3), increase by 4.1 % (rice cropping) to 40.3 % (bare soil). The spatial pattern of P–E change in different Amazonian land use scenarios (bare soil, dry pasturelands, wet pastureland, and rice cropping) can be seen in Fig. 6. As it shows, generally, land use scenarios for almost the entire Amazon basin results in

a surface runoff increase across the Amazon basin but a decrease outside of it. Runoff increase within the Amazon basin also shows the spatial differences as it is more pronounced in the north-eastern part of the Amazon and less significant in the western part.

.



Similarly as in our investigation on smaller domains such as the MIP of the target sink, we apply different land use scenarios in the domains of Amazon basin and the Amazon basin without the Ucayali river basin to investigate the upper bounds of the rainfall and runoff reception changes in the target sink, the Ucayali River basin outlet (see Table 6). The comparison of these upper bounds with the impacts from the influential sources hotspots are presented in Sect. 4.3. Rainfall

in the target sink decreases by 10 % (rice cropping) to 26 % (bare soil) in all cultivated Amazon basin scenarios, but runoff in the target sink increases by 11 % (wet pastureland) to 33 % (bare soil). Converting the whole Amazon basin into rice cropping has in fact very small influence on the runoff received by the target sink (−1 %). Contrary to the results from applying scenarios to the Amazon basin, runoff decreases by 27 % (rice cropping) to 69 % (bare soil) when applying land use scenarios in the domain of the Amazon basin without the Ucayali river basin. This resonates with the findings in Sect.

3.1 that applying land use scenarios out of the watershed boundary has negative influences on the runoff of the target sink and is discussed in the following section.

## 4 Discussion

### 4.1 Sensitive sinks under pressure

The sensitive areas most dependent on the moisture recycled from the Amazon forest have been identified as being

situated in the Peruvian Amazon and its transition to the Andes, such as the Junín, Cusco, Puno regions, and a part of western Bolivia. Given that the average annual rainfall of the sensitive areas is 997 mm yr$^{-1}$, the 11.3–38.5 % rainfall reduction from the upper bound of our investigation has considerable impacts on the ecosystems and agriculture in those areas, especially during dry seasons (Bagley et al., 2014, Alves et al., 2017). Though this upper limit in which land use change takes place in the whole Amazon is hypothetical, land use change within the MIP, covering 3.5 % of the Amazon, is

possible (Aguiar et al., 2016). As it controls half the Amazonian provision of evapotranspiration in the sensitive areas, the land use change taking place in the MIP has a greater ability to alter the rainfall of the sensitive regions compared with that occurs in the rest of the Amazon basin. The location of the MIP for the sensitive areas is here identified in the Ucayali and Madre de Dios region of Peru, as shown in Fig. 3. About 2.76 % of the forests were cleared in the Ucayali region in the period between 2001–2014 (MINAM, 2017) but the deforestation rate is expected to increase because of continuing

migration into these regions and increasing investment in roads and transportation (Piu and Menton, 2014).

Our results on the spatial pattern of rainfall dependency in the Amazon basin (Fig. 2) agree with maps produced in studies on other aspects of moisture recycling (Eltahir and Bras, 1994, Figs 4 and 6; Burde et al, 2006) though the recycling ratios may be a slight overestimation along the Andes because of the imbalance between the input precipitation TRMM product and the evapotranspiration product MOD16ET. Nevertheless, the overestimation is small when the MOD experiment

reports general agreement with other studies using other datasets (see Table 2; Zemp et al., 2014).





## 4.2 Interconnected aerial and surface rivers – spatially different response to land use change

Our investigation suggests that the sensitive areas' rainfall reacts more significantly to land use change in the Amazon basin, by doubling the average rainfall reduction of the Amazon basin and tripling that of the South American continent average, and this propagates to the runoff responses in the sensitive areas. Taking the upper bound investigation for instance,

significant drops in evapotranspiration due to land use scenarios applied within the Amazon basin lead to higher runoff estimates (P–E surpluses) throughout the basin. However, these runoff rises are more compensated in sensitive catchments which experience more significant rainfall reduction by land use change. This is reflected in the spatial heterogeneity in the extent of runoff response across basins (Fig. 6, also see Table 5 for the comparison between the Ucayali river basin and the Madeira river basin runoff responses). As shown in Fig. 6, the rise in P–E in each scenario becomes less prominent towards

the western Amazon, corresponding to growing sensitivity of the rainfall to Amazonian land use change (see Fig. 2). The north-east Amazon, where rainfall is the least dependent on Amazonian evapotranspiration, reports the greatest growth in the P–E surplus in all scenarios.

We estimated altered rainfall by land use change through the moisture recycling process while neglecting the moisture pathway dynamic resulting from the altered energy balance (Shukla et al., 1990; Bonan et al., 2008, Mahmood et al., 2014;

Lejeune et al., 2015), deepening convective boundary layer (Fisch et al., 2004) and reduction in surface roughness (Khanna and Medvigy, 2014) after land use change. Nevertheless, our estimate of shifts in rainfall by land use change is in line with results from studies considering such effects. Our calculation of an annual rainfall reduction of 10.4–12.7 % in both wet and dry pastureland Amazon scenarios falls in the range of a mean 16.5 ± 13 % reduction in annual rainfall of the Amazon basin, reported from 44 GCM and RCM studies that hypothetically converting 100 % of the Amazon into soybean or pastureland

use in Spracken and Garcia-Carreras's meta-analysis (2015). 18 out of the 44 studies also considered roughness and albedo changes through coupled runs with land surface models or biosphere models. Our estimates are still in agreement with their results reporting an average 15.3 ± 8% reduction in annual Amazon rainfall (Spracklen and Garcia-Carreras, 2015). In this case, the neglected processes have minor influences on our overall results.

As for runoff discharges, modelling outputs from previous studies applying Amazon deforestation scenarios have

diverse predictions. Some report increments after land use change (Dirmeyer and Shukla, 1994; Lean and Rowntree, 1997; Kleidon and Heimann, 2000) and some found a decrease (Henderson-Sellers et al., 1993, Hahmann and Dickinson, 1997, Voldoire and Royer, 2004). Our results show that runoff response differs from basin to basin and depends on alternative land use practices. This spatial heterogeneity in the P–E response (as shown in Fig. 6) may contribute to the diversity of the findings from previous studies.



### 4.3 Water conservation hotspots out of watersheds

Our results suggest that a given region's water availability is not only related to land activities in its upstream watershed but is highly controlled by those in its MIP and its watershed's MIP. These are the areas not necessarily located in the upstream watershed which is traditionally considered in the land use assessments for water conservation.

The importance of land use change in the MIP on the target sink's rainfall is shown by comparing with impacts on rainfall by land use change in the whole Amazon (the upper bound investigation in Sect. 3.2). In our exploration for the Ucayali River basin outlet as a target sink, land use change in its MIP results in a 5–12 % drop of the target sink's rainfall. This is considerable compared with a 10–26 % decrease in the target sink's rainfall by land use change in the whole Amazon basin, 9 times the size of the target sink's MIP. By contrast, when land use change occurs in the Ucayali river basin, the

reduction in the target sink's rainfall is considerably lower (by less than 5 %).

The interconnection between surface rivers and aerial rivers implies that the land use changes taking place out of the watershed can be crucial to the runoff reception. In fact, in our investigation, land use change that happens in the target sink's upstream watershed brings converse impacts on runoff compared with land use change taking place out of the target sink's upstream watershed. We found an abundant increase in the runoff received in the Ucayali river outlet, the target sink,

when land use scenarios are applied in the Ucayali basin. This is consistent with modelling and observational studies that investigate runoff response to land use change in a specific sub-basin or catchment (Costa et al., 2003; Coe et al., 2011, Panday et al., 2015). However, the runoff reduces by 27–69 % when employing land use scenarios in the domain of the Amazon basin excluding the Ucayali river basin (see Table 6). Within this area, land use change in the MIP of the watershed is more influential on the target sink's runoff. The results is a 19–50 % reduction, even though its areal content is less than

half that of the Amazon basin excluding the Ucayali basin. This also reflects that when applying land use scenarios at a pan-Amazon scale, runoff estimates of a specific watershed yield contradicting responses to land use change in different moisture source areas (within that watershed-positive, outside of that watershed-negative).

### 4.4 Managing interconnected surface and aerial rivers crossing boundary

Our results suggest that sensitive sinks (eg. the sensitive areas, see Sect. 3.1) and influential sources such as the MIP of

the given region and the MIP of its watershed are those areas crucial for managing water availability under interconnected aerial and surface river regimes. In order to do this, transboundary involvement crossing regions, municipalities, provinces or countries is necessary. For example, our results of the sensitive pairs reflect that as it is located in the downstream area of the aerial river, the Bolivian sensitive areas should recognise the importance of the land activities in the neighbouring Peruvian Amazon. For another example of the target sink in the Ucayali basin outlet, though its watershed area is located completely

in Peru, its MIP has Peruvian, Brazilian and Colombian components. Therefore, for the Amazon countries' sustainable use and management of the fresh water, understanding the roles in the aerial river regime within and across individual countries and initiation of co-management are crucial. Previously, Dirmeyer et al., (2009) has investigated the imports and exports of



the moisture per country globally. Though these moisture budgets can be useful for understanding each country's dependency, they provide limited spatial information for conservation targets. Keys et al. (2012) have introduced the concept of precipitationsheds to identify areas providing moisture for precipitation in downwind areas. Our results further show that a particular component of the precipitationshed with small areal extent can be especially influential (MIP) for rainfall and that

the interlinkage between the aerial and surface rivers marks the importance of the MIP of the watershed on runoff. The identification of such hotspots and quantification of potential hydrological influences by land use change within them provides conservation targets for sustainable management of interconnected surface and aerial river regimes crossing boundaries.

## 4.5 Limitations

Our analysis based on the average output of period 2000–2010 from the MOD experiment has not accounted for the interannual variation of moisture recycling, though it is generally reported as small in the Amazon basin (Bosivolich and Chern, 2006). However, we note that the two major droughts (2005 and 2010) in the simulated period of the MOD experiment may lead to an over-estimation of the moisture recycling influence (Bagley et al., 2014). The seasonal variation was also masked despite the slight difference (3–5 %) reported by Zemp et al. (2014) between dry and wet seasons in the

precipitation recycling ratio in Amazonia over the investigation period. We are aware that the spatial patterns of recycling varying through the seasons (Arraut et al., 2012; Zemp et al., 2014) and that can influence the identification of the MIP location. However, Keys et al. (2014) concluded that the core-part of precipitationsheds can be suggestive for the analysis of terrestrial precipitation recycling, which may be reflected by our decadal average results. Still, further studies that focus on seasonal specific purposes such as rain-fed agriculture should take the growing season's precipitationshed shift into account.

Other uncertainties could remain in the extrapolation of LBA-ECO flux tower data measured in the Santarém region for the entire Amazon basin. The spatial variability in evapotranspiration that might arise from varying environmental conditions (Fisher et al., 2008) is not considered. However, the evapotranspiration approximation is still site and ecoregion based (Christoffersen et al., 2014) while the evapotranspiration modelling power over Amazon forest ecosystems is still poor (Karam and Bras, 2008; Werth and Avissar, 2004, Maeda et al., 2017).

In the present study, we focus on land use change's effect on moisture availability through the moisture recycling process. Other processes are also known to be involved in shifting water regime when land use change occurs; for example, rising aerosols modifying cloud microphysics (Koren et al., 2012), changed surface roughness (Khanna and Medvigy, 2014; Khanna et al., 2017), and its forcing on convective systems (Baidya Roy and Avissar, 2000; Baidya Roy, 2002; D'Almeida et al., 2006). Feedback mechanisms such as vegetation–atmosphere interaction intensifying droughts and driving large forest

die-back (Nepstad et al., 2008; Malhi et al., 2009; Zemp et al., 2017) can also influence the rainfall and runoff regime. Since our study has suggested the sensitive sinks and influential sources' importance on the shifts in water regime, further studies on how these processes interact with moisture recycling spatial heterogeneity can further advance our insights into the water regime shifts by land use change.





## 5 Conclusion and outlook

From our analysis of the moisture recycling process, we conclude that Amazonian land use change's impacts on the water regime have spatial heterogeneity in two ways. First, hydrological responses in moisture sinks vary spatially. Second, land use change in different locations exerts varying influences. Under this spatial heterogeneity, sensitive sinks and

influential sources can be identified. Using a moisture tracking experiment of a water balance model (WAM-2layers), we have identified the sensitive areas to Amazonian land use change in the semi-arid southern Peru and eastern Bolivia. We quantified changes in rainfall and runoff by various land use scenarios in the Amazon and found sensitive areas experience more significant rainfall reduction (11.3–38.5 %, depending on scenarios) and a lower runoff increase (0.9–32.7 % in the Ucayali river, depending on scenarios). In addition, we have introduced the concept of MIP (most influential

precipitationshed) where the most important source areas of moisture for a given region collectively situate and back-tracked the MIP of the sensitive areas, which is located in the Ucayali and Madre de Dios region in Peruvian Amazon. By exploring land use change's varying influences on a target sink's water availability from different source areas, we found that land use change in the upstream watershed of the target sink leads to a runoff rise. However it can also lead to a reduction when land use change occurs out of its upstream watershed. We also identified the MIP of the target sink's upstream watershed as the

hotspot for conserving runoff (19–50 % reduction, depending on scenarios) and MIP of the target sink as the hotspot to conserve rainfall (5–12 % reduction, depending on scenarios) for land use assessment.

Our analysis on the importance of spatially different land use change impacts on the water regime can explain the diversity of other modelling experiments findings. Macro-scale experiments reflect aggregated influences and responses from different spatial components, thus they do not contradict different findings from mesoscale experiments, in which

estimates are geographically specific. Nevertheless, for conservation targets, these aggregated results are rarely suggestive. For future meso-scale analysis, we suggest a shift of spatial focus from a pure watershed study because land use changes out of a target area's watershed can also be very influential. Our results also reflect that the deforestation tipping point beyond which rainfall changes will lead to strong rainfall reductions with drastic ecological impact on the forest found in Lawrence and Vandecar (2015) can be lower when the deforestation takes place in influential source areas, such as MIPs.

At a national level, we suggest that a crucial step towards the Amazon countries' sustainable usage of water (resonating the fulfilment of SDGs 6 and 15) is to include the influence of land activity in water management. However, other than traditionally recognising only upstream watershed regions in the water management, land use in the precipitationsheds, especially the MIPs, is of importance in both the rainfall and runoff regime sustaining the ecosystem (Coe et al., 2013) and agriculture (Bagley et al., 2012; Keys et al., 2014). Our results also highlight the importance of

transboundary cooperation along both the surface and the aerial river for managing water regime shift by land use change. Top-down international laws and regulation offer an opportunity (Keys et al., 2017) but bottom up national efforts should focus on understanding each country's role in the aerial river regime crossing boundaries and the places in need for action. It can be done by recognizing the moisture sinks sensitive to land use change and locating influential sources (MIPs) that exert




strong controls on the rainfall and runoff regime of the sensitives. Their importance and the methods to identify them were demonstrated in this study.

*Code availability.*

The WAM-2layers model code is available on https://github.com/ruudvdent/WAM2layersPython under the GNU General Public License.

*Data availability.*

The LBA-ECO flux tower data is available online at https://daac.ornl.gov/cgi-
bin/dataset_lister.pl?p=11#surf_hydro_and_water_chem_anchor and Sakai et al. (2004).

*Competing interest.*

The authors declare that they have no conflict of interest.

*Acknowledgements.*

This research was supported by the German International Climate Protection Initiative (project: Sustainable Latin America, reference number 42206-6157). D.C.Z. acknowledges funding from the DFG (project IRT1740). We thank F. Gollnow, S. L. Becker and D. M. Landholm Haight for comments on this manuscript. We also thank R. van der Ent, K. Thonicke and A. Rammig for discussions on the WAM-2layers used in this study.

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



**Table 1. The specification of the MOD experiments which was used in our analysis to trace the moisture**

| Specification of the MOD experiments | |
| --- | --- |
| Precipitation input | Tropical Rainfall Measuring Mission (TRMM) Multisatellite Precipitation Analysis (TMPA) |
| Evapotranspiration input | Moderate Resolution Imaging Spectroradiometer (MODIS) product MOD16ET |
| Humidity and wind speeds | ERA-Interim reanalysis |
| Temporal resolution | 3 h |
| Spatial resolution | 1.5°×1.5° |
| Experiment time span | 2000–2010 |
| Spatial domain | South American continent (land part of 30° W–85.5° W, 15° N–49.5° S) |



**Table 2. LBA-ECO evaporation data**

| Land use type | Bare soil | Dry pastureland | Wet pastureland | Rice cropping |
|---|---|---|---|---|
| Evaporation rate (mm d$^{-1}$) | 1.2±0.7 | 1.9±0.6 | 2.2±0.9 | 2.7±1.2 |

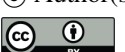



**Table 3. Estimated changes in annual rainfall (ΔP) and runoff (ΔR) over the Ucayali River basin outlet following land use scenarios in different spatial domains.**

| Land use change domain | Ucayali basin | | MIP of Ucayali river outlet | | MIP of the river basin excluding the Ucayali basin | |
|---|---|---|---|---|---|---|
| Ucayali river outlet's water regime | ΔP | ΔR | ΔP | ΔR | ΔP | ΔR |
| Bare soil | –3 % | +103 % | –12 % | –9 % | –16 % | –50 % |
| Dry pastureland | –2 % | +67 % | –8 % | –7 % | –12 % | –36 % |
| Wet pastureland | –2 % | +52 % | –7 % | –6 % | –10 % | –30 % |
| Rice cropping | –1 % | +27 % | –5 % | –4 % | –7 % | –19 % |

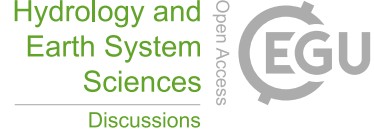

**Table 4. Estimated changes in annual rainfall over different regions when applying various land use scenarios in the Amazon basin. Note that annual rainfall is reduced continental wise, but the sensitive areas experience greater reductions.**

| | Area (km²) | Rainfall dependency on the Amazon basin (%) | Rainfall change for different land uses (%) | | | |
|---|---|---|---|---|---|---|
| | | | Bare soil | Dry pastureland | Wet pastureland | Rice cropping |
| Sensitive areas | $3.25 \times 10^5$ | 60.3 | −38.5 | −25.8 | −20.4 | −11.3 |
| Amazon basin | $7.77 \times 10^6$ | 27.5 | −18.2 | −12.7 | −10.4 | −6.5 |
| South American continent | $1.70 \times 10^7$ | 20.0 | −12.9 | −8.8 | −7.0 | −4.0 |





**Table 5. Runoff (P–E) estimates in different regions under different land use scenarios.**

|  |  | Control | Bare soil | Dry pastureland | Wet pastureland | Rice cropping |
|---|---|---|---|---|---|---|
| Ucayali basin (3.1 % of the Amazon) | P–E in the Ucayali basin (10 km$^3$ yr$^{-1}$) | 23.285 | 30.891 | 27.444 | 25.966 | 23.504 |
|  | Comparison with the control group | – | +32.7 % | +17.9 % | +11.5 % | +0.9 % |
| Madeira basin (13.9 % of the Amazon) | P–E in the Madeira basin (10 km$^3$ yr$^{-1}$) | 103.15 | 144.68 | 127.39 | 119.84 | 107.42 |
|  | Comparison with the control group | – | +40.3 % | +23.4 % | +16.2 % | +4.1 % |



**Table 6. Estimated changes annual rainfall (ΔP) and runoff (ΔR) over the Ucayali River basin outlet following land use scenarios in the Amazon basin and the Amazon basin excluding the Ucayali basin.**

| Land use change domain | Amazon basin excluding the Ucayali basin | | Amazon basin | |
|---|---|---|---|---|
| Ucayali river outlet's water regime | ΔP | ΔR | ΔP | ΔR |
| Bare soil | −23 % | −69 % | −26 % | +33 % |
| Dry pastureland | −17 % | −49 % | −19 % | +18 % |
| Wet pastureland | −14 % | −41 % | −16 % | +11 % |
| Rice cropping | −9 % | −27 % | −10 % | −1 % |





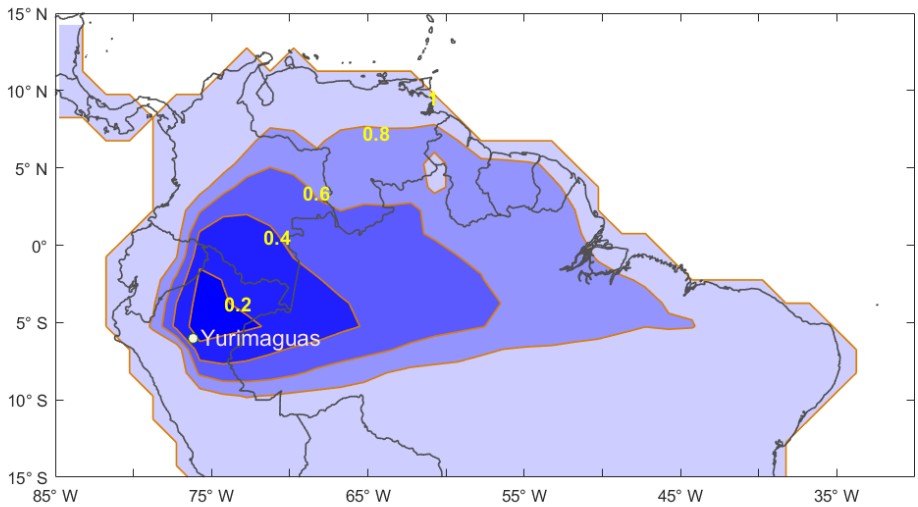

**Figure 1. The precipitationshed of the Yurimaguas area. The contour value represents the cumulative fraction of Yurimaguas's rainfall that comes from the source region delimited by the contour, over the precipitation originating from the South American continent. Thus, the contour line delimiting the South American continent has the value 1.**



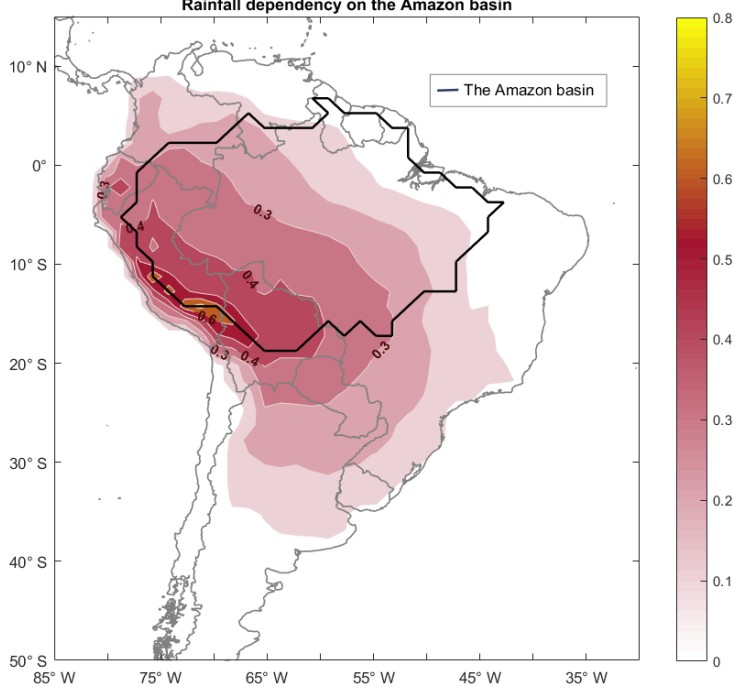

**Figure 2. Rainfall dependency on the Amazon basin. The number shows the fraction of local rainfall recycled from Amazonian evapotranspiration. The yellow areas are among those regions having the greatest sensitivity to Amazonian land use change.**





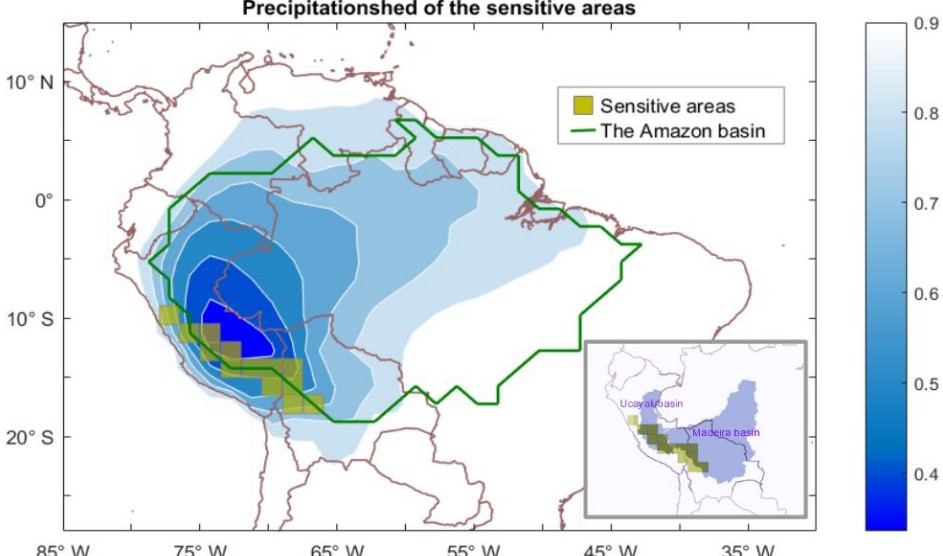

**Figure 3. The precipitationshed of the sensitive areas. The contour value stands for the fraction of rainfall from continental evapotranspiration in sensitive areas that is evapotranspirated from the delimited region collectively. The first contour delimits areas (shown in dark blue) corresponding to the most influential precipitationshed (MIP) for the sensitive regions (represented by yellow cells). 74.7 % of the sensitive areas' total rainfall comes from continental evapotranspiration. Among this, 40 % originates from the MIP.**





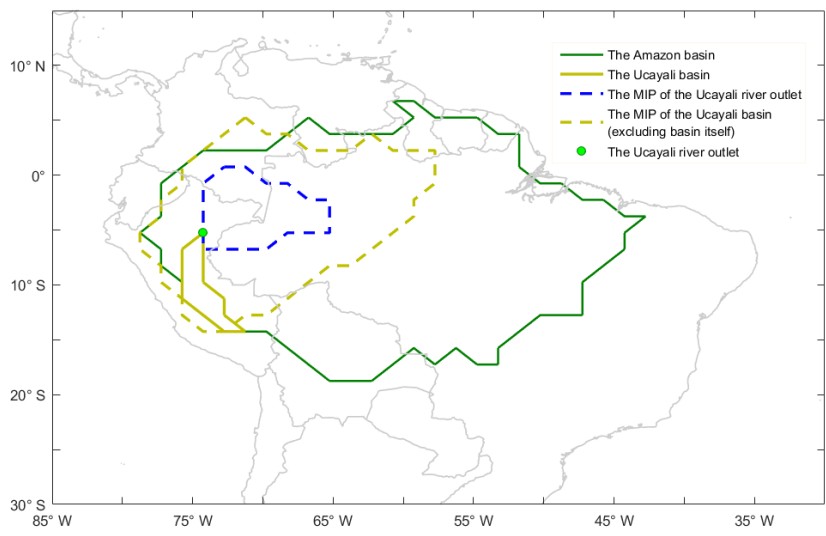

**Figure 4. Different land use scenarios domains for exploring rainfall and runoff susceptibility of the target sink (Ucayali River outlet). These domains include the Ucayali River basin (the watershed of the target sink), the MIP of the target sink and the MIP of the Ucayali river basin (the MIP of the watershed) but excluding the basin component cells, in order to understand land use change influences outside of the watershed boundary. In addition, land use scenarios are also applied in the domains of the Amazon basin and the Amazon basin without the Ucayali river basin for upper bound investigation.**

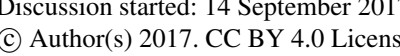



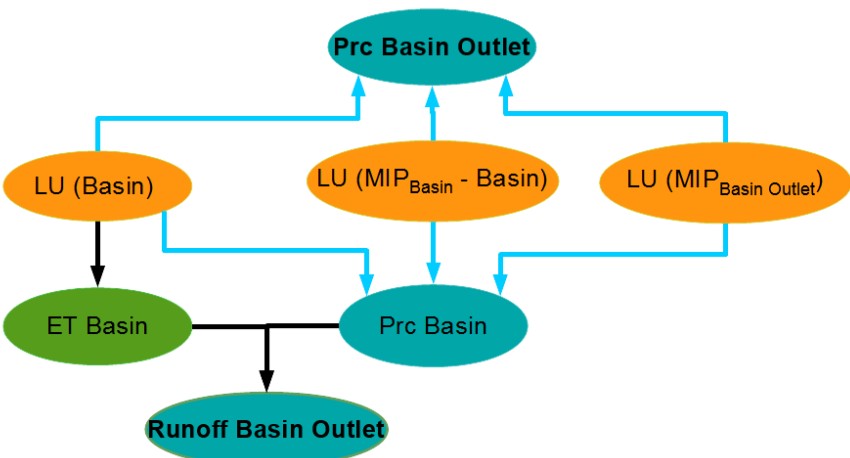

**Figure 5. Influence of land use (LU) in different spatial domains (orange ellipses) on runoff and precipitation (cyan blue ellipses) of the outlet of the basin. Light blue arrows show influences via moisture recycling ("aerial rivers"), black arrows represent surface-bound relations. ET denotes the annual evapotranspiration of the basin and Prc stands for precipitation.**



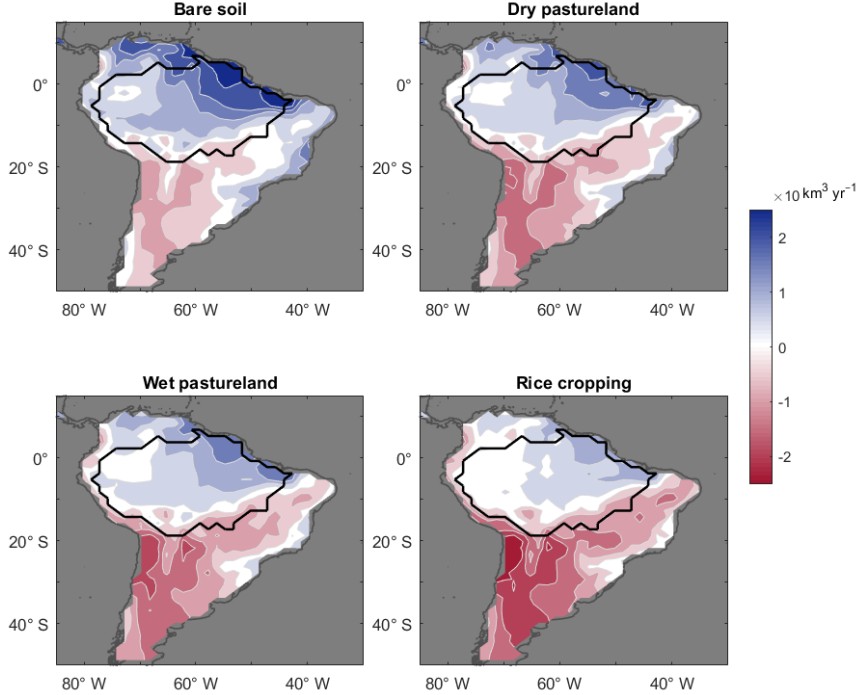

**Figure 6.** Spatial patterns in local runoff (P–E) changes compared to the control state for land use scenarios applied in the Amazon basin. Runoff generally increases in all scenarios (especially in the north-eastern part of the Amazon basin) but the rise is less pronounced in the rice cropping scenario over Amazonia