# Peer review of "Aerial and surface rivers: downwind impacts on water availability from land use changes in Amazonia"

_Hydrology and Earth System Sciences, 2017_

## Referee Comment (RC1) · P.W. Keys (Referee) · 17 Oct 2017

GENERAL COMMENTS

Weng et al. (hereafter, the authors) explore the role of land-use change on the hydrology of the Amazon, focusing on the implications of changes in evaporation on moisture recycling, precipitation, and subsequent runoff. The authors identify key regions in the Amazon that are particularly sensitive to changes in continental moisture recycling, and further identify how different land-use change scenarios can impact hydrology.

I find the study to be interesting, relevant, and timely. I recommend the paper receive minor revisions, prior to acceptance for publication in HESS. I will summarize my comments briefly, and then lay out more detailed comments below.

[Figure]

1) The paper is on the cutting edge of land-use change & moisture recycling research, in the sense that it is examining sensitive regions to land-use change, the role of internal-watershed vs. external-watershed land-use change impacts, and the importance of different types of land-use change having very different types of consequences on hydrology. I encourage the authors to emphasize the 'cutting-edge' nature of their work a bit more, and be more bold in their conclusions.

2) I'm very interested in the development of the 'most influential precipitationshed' (MIP) concept. However, the MIP is essentially a definition of a precipitationshed boundary. The precipitationshed boundary that contains 100% of evaporation is the entire planet (at least in the context of the WAM-2layers). Thus, anything less than that domain requires the selection of a boundary. There is quite a bit of discussion on boundary selection and comparative advantages and disadvantages in the existing literature (e.g. the 70% boundary in Keys et al. (2012), the identification of the 'core precipitationshed' as a persistent, inter-annual source of moisture in Keys et al. (2014), and the discussion of the 1% boundary in Keys et al. (2017)). It is good that the authors are innovating on the concept of the precipitationshed, but I think the MIP should be put into better context as an approach for quantifying a boundary.

3) Regarding the MIP, I think that the authors need to emphasize more clearly that the 40% threshold is related to grid resolution (as far as I can tell, the only reference to this is at line 22-24).

Again, I think this paper is quite good, and is in need only of minor revisions before publication.

DETAILED COMMENTS (P = Page, L = Line)

P2 L4 This sentence is confusing, especially the section "...operate under uncertainties of the undergoing land use change...". Please revise.

P2 L24-25 I suggest the authors remove the part of the sentence "...,which has not

been covered in depth by previous studies", since many studies have looked in detail at how land-use change might impact the hydrological regime in the Amazon. There is still much work to do of course, but there has still been quite a lot of research into land-use change, moisture recycling and the hydrological cycle throughout the Amazon.

P2 L22 Consider including Badger & Dirmeyer (2015) and Keys et al. (2016). Badger and Dirmeyer conducted a detailed examination of the climate impacts of land-use change in the Amazon, including a very detailed analysis of the hydroclimate. Keys et al. examine the role of vegetation change on moisture recycling, including a regional focus on a part of the Amazon experiencing rapid land-use change (as well as using the WAM-2layers for the moisture tracking).

P2 L26 If this is the first instance of the abbreviation 'SDGs', please spell it out. Also, which of the 17 SDGs are the authors referring to? Consider adding some specificity here and a citation to support the relevance of moisture recycling (I'm not doubting its relevance, but it would be useful for the authors to chart this relevance more clearly and specifically).

P5 L4 In section 2.1.3, the authors explain their concept of the 'most influential pre-cipitationshed'. From my understanding, this is simply a threshold-based boundary, correct? As stated earlier, this a very interesting idea, but the authors ought to acknowl-edge that this is one of several methods for delineating a precipitationshed boundary. I highlighted the previous studies in my General Comments that have discussed bound-ary methods. Essentially, the 40% MIP is the boundary which provides 40% of conti-nentally recycled precipitation, correct?

P6 L26 It would be useful to remind the reader that MIP essentially means the 40% terrestrial moisture recycling boundary (again, assuming I understand it correctly).

P7 L12 What is meant by "with high spatial efficiency"?

P7 L13 Why does the MIP account for 50% of the Amazonian evapotranspiration?

Shouldn't it be 40%? Please clarify for easier interpretation of the result.

P8 L7 A bit confusing. Please replace "adding an extra time. . . on the original flow" with "more than doubling. . . the original flow"

P8 L22 Perhaps replace "fashion" with "pattern"?

P8 L30 Very interesting finding!

P9 L6-7 The finding about the rice planting not having very large impacts on run-off makes me curious about seasonal impacts (e.g. trees evaporate at different times than crops, etc.). Did you explore seasonal impacts? If so, please include some information on that analysis; if not, please include a few comments as to why it is outside the scope of this present work.

P9 L20 "As it controls half the Amazonian evapotranspiration. . ." Again, I am confused about whether the MIP represents 50% or 40%.

P10 L25 Do the authors mean "increases" where they wrote "increments"?

P11 L12 The authors should consider citing Wang-Erlandsson et al. (2017) since they find these same types of results. Both the Wang-Erlandsson paper and this paper are currently in HESSD, and it would be useful as a reader to see they find complimentary results using a variation in methods. In the interest of full disclosure, I am a co-author on the Wang-Erlandsson et al. article, and will suggest to the lead author of that paper that they also cite this work (for the same reasons I suggested already).

P13 L2-5 Here the authors could be bolder in their conclusions about what is important and novel about their work. E.g. The importance of relatively small source areas for sensitive regions in the Amazon; also the importance of extra-basin land-use change on basin runoff.

P14 L1 Confusing sentence ". . .strong controls on the rainfall and runoff regimes of the sensitives." I think the authors are missing a word; perhaps "sensitive regions"?

P14 L1-2 I recommend removing the final sentence since it is unnecessary.

Fig 2 & 3 Both figures need a label on the colorbar

REFERENCES Badger, A. M. and Dirmeyer, P. A.: Climate response to Amazon forest replacement by heterogeneous crop cover, Hydrol. Earth Syst. Sci., 19, 4547-4557, https://doi.org/10.5194/hess-19-4547-2015, 2015.

Keys, P. W., Wang-Erlandsson, L., & Gordon, L. J. (2016). Revealing invisible water: moisture recycling as an ecosystem service. PloS one, 11(3), e0151993.

Wang-Erlandsson, L., Fetzer, I., Keys, P. W., van der Ent, R. J., Savenije, H. H. G., and Gordon, L. J.: Remote land use impacts on river flows through atmospheric teleconnections, Hydrol. Earth Syst. Sci. Discuss., https://doi.org/10.5194/hess-2017-494, in review, 2017.

---

## Referee Comment (RC2) · R.S. Westerhoff (Referee) · 19 Oct 2017

This paper addresses the role of soil moisture recycling (aerial rivers) in the water balance, and how it affects downwind climates in scenarios of land-use, more specifically forest change in the Amazon. The paper reads well and touches on a topic that is not often discussed amongs hydrologist. The study also potentially has a large impact and therefore much relevance. I have to so that I am not an expert in soil moisture recycling and I could therefore not go into detail on validity of all the methods used. However, I do see that the paper could use some extra description and discussion on uncertainty.

The only major comment I have is that the figures that you present incorporate some uncertainty or range, but without any explanation on how these were estimates. Also,

other uncertainties are not addressed, e.g., the assumption of constant groundwater. Some uncertainties might be higher than your actual estimates. Or not, but without proper explanation we do not know. I think the paper also deserves a discussion that deals with uncertainty.

Since my comments are relatively easy to address (in my opinion) I therefore recommend this paper to be accepted with minor revisions.

These are my comments. Please treat the comments on uncertainty as less than minor.

Page 2, line 7. Two sentences that almost say the same, try turning these into one.

Page 2, line 19-21. The sentence is unclear, especially the part "some areas' water regime". Please rephrase.

Page 2, line 26. Explain the abbreviation SDGs. Maybe a reference to some of these SDGs (e.g. water)

Page 2, line 34. The first time you use the term aerial rivers, explain what they are. That is important, since the term is in the title. You can either probably solve that quite simple by saying:" 'aerial rivers', i.e., preferential pathway of moisture flow, termed in Arraut et al. (2012) as an analogy to surface rivers."

Page 3, line 6-10. Try to avoid method description in the introduction.

Page 3. Define what the sinks and sources are in this study.

Page 5, line 22. It is e.g., not eg.

Page 6, line 13. I think you should explain the assumption of Zemp et al. (2014) in somewhat more detail, instead of the quick reference. What is balance? Are they equal? Or are their ratios equal?

Page 6, line 13. The steady groundwater storage assumption is of major importance in my opinion. This needs to be in the discussion. For example, removing trees generally

elevates the water table. Although the water table is already very shallow in most of the Amazon, small differences of e.g. 5 cm might have mahor differences in the whole balance that you are calculating. Can you something on the uncertainty surrounding that?

Page 8, line 6-7. a quarter of what, and extra time of what?. This sentence should be much clearer.

Page 9, I would like to see some more uncertainty discussed. E.g. the groundwater assumptions. Also, it is not entirely clear from the method how you got your result uncertainty ranges (e.g., the 10-26%m 5-12% etc).

Page 9-10: Discussion: Can the discussion mention what the relative contribution is compared to the moisture from the sea?

Page 13, line 31. It is 'bottom-up'

---

## Short Comment (SC1) · 7 Nov 2017

Note to the editor and authors: As part of an introductory course to the Master pro-gramme Earth & Environment at Wageningen University, students get the assignment to review a scientific paper. Since several years, students have been reviewing papers that are in open online discussion for HESS, and they have been asked to submit their reports to the discussion in order to help the review process. While these reports are written as official (invited) reviews, they were not requested for by the editor, and we leave it up to the editor and authors to use these reports to their advantage. While several students were asked to review the same paper, this was not done with the aim to provide the authors with much extra work. We hope that these reports will positively contribute to the scientific discussion and to the quality of papers published in HESS.

[Figure]

This report/review was supervised by dr. Ryan Teuling.

INTRODUCTION

The paper discusses the impact of land use change on the hydrology of the Amazon. Land use change is happening a lot in the amazon these days and therefore, it is important to know the impact of this change on the regional and global scale. By using a moisture recycling tracking algorithm, Weng et al. tried to get a better understanding of the influence of land use change on rainfall, evapotranspiration and runoff. The results of the paper are that by all the land use changes the rainfall decreases. The extend of this change depends on the location where the land use changes happen. Furthermore, a change of the whole Amazon to a certain type of land use has a large influence on both the precipitation and the runoff.

The paper touches a topic which is very relevant at the moment, other studies have been looking into this as well(Snyder, 2010, Gordon et al., 2005), but the spatial different sensitivity in the hydrological responses to land use change was not well understood. Like said in the paper itself, deforestation is happening in the Amazon to create agricultural land(INPE, 2017). This change to agricultural land use can have a massive impact on the hydrology in the Amazon. Due to the fact that the Amazon is such a large area, this could even have an influence on the world as well. Therefore, it is a very interesting topic which should be looked into even more with other researches.

The paper is well written, but there are some minor improvements which could be made to make this better. These minor improvements will be stated later on in this review. Little research is done at this topic so the research which is conducted by Weng et al. is innovative. It is interesting because the outcomes of this research can be used in other areas which suffer from deforestation as well. For this reason this paper can be an eye opener for other people to investigate this process even more. The hydrological impact which is the result of this paper perfectly fits the aim of the Hydrology and earth system sciences (HESS) scientific journal. The research which is conducted is done

well, but there are some minor issues which could be solved. Therefore, I recommend some minor revisions before publications by HESS. The revisions that should be made in my opinion are listed below

GENERAL COMMENTS

In my opinion there should be made a better substantiation of the use of MOD16ET data. The authors say: "Loarie et al. (2011) validated MOD16ET's estimation with eddy flux tower data and reported its good performance (differences in annual average of evapotranspiration are less than 4 % in savannas, 5 % in tropical forests and 13 % in pasture agricultural lands)". However, other references say something else: "While all three evaporation products adequately represent the expected average geographical patterns and seasonality, there is a tendency in PM-MOD to underestimate the flux in the tropics and subtropics. Overall, results from GLEAM and PT-JPL appear more realistic when compared to surface water balances from 837 globally distributed catchments and to separate evaporation estimates from ERAInterim and the model tree ensemble (MTE)."(Miralles et al., 2016). These references are opposites of each other. The use of the MOD16ET method can have an uncertainty on all the figures and results that are made in this research.

I would suggest to make a better substantiation why the MOD16ET data is used and why for example the GLEAM or the PT-JPL were not used. Furthermore, a paragraph can be added to the discussion with the topic what the uncertainty of the MOD16ET is on the results that are made.

A second revision is to give more substantiation and discussion on the use of the WAM-2layers model. The WAM-2layers simulations of another experiment are used but the use of a WAM-2layers offline model give worse results than an online model like the RCM-tag model. In a paper by Van der Ent et al., 2013 a comparison is made between the WAM-2layer model and the RCM-tag model, a result of this comparison was that simulations of both models give globally the same result. However, at a regional scale,

the error for the recycling ratio of the WAM-2layers model is relatively large if it is compared with this error of the RCM-tag model(respectively 2.8% against 1.9%(Van der Ent et al., 2013)). The research is mostly about the Amazon, which is a regional scale as well. Therefore, the results and figures could be different when a more precise method was used. I suggest the authors to take this into account in the discussion as well. The use of the WAM-2layers model has a larger uncertainty than an online model. Therefore, this uncertainty should be mentioned in the discussion.

A third revision is the title of the manuscript is: "Aerial and surface rivers: downwind impacts on water availability from land use changes in Amazonia". This gives the feeling that the paper is about the water availability in the downwind areas. However, the conclusions that are stated in this manuscript are all about discharge and reduction of precipitation. If I look at the definition of water availability in a random dictionary I get the following: "The portion of a water resource that can be abstracted, as determined by the total water resource and the rights to abstract water from that water resource.". So the title will attract readers who are interested in the amount of water which is available in the amazon to abstract. The first sentence of the conclusion is: "From our analysis of the moisture recycling process, we conclude that Amazonian land use change's impacts on the water regime have spatial heterogeneity in two ways.". So the conclusion is about the water regime, not the availability. I would suggest that the title of the manuscript is going to be changed, especially the term "availability". This is a term which could attract the wrong readers. In my opinion, a term like "water regime" would fit better in this manuscript.

MINOR COMMENTS

P2, line 7: place a space between 80 and the % sign

P2, line 26: There is an abbreviation SDGs in this sentence but it is not said what this abbreviation means.

P4, line 6-10: this is part of a methodology already. This should not be in the introduction

P5, line 22: eg. Should be e.g.

P7, line 25 and 26: in the first line the reference to a figure is like: "Fig. 4" in the next sentence the reference is like: "figure. 5" Please be consistent. Use "Fig" or "Figure" but not both

P8, line 15: "For that, we apply in the following..." remove "in".

P11, line 19: "The results is...." Remove the "s" in the word "results".

Fig 1: Has no title in the figure itself.

Fig 2: What are the units of the color bar? Give it a label.

Fig 3: What are the units of the color bar? Give it a label.

Fig 4: Add a title at the figure itself

Fig 6: Try to give it the same mask as the other figures, now the whole of south America is showed while the results that are mentioned are only about the amazon.

REFERENCES

Gordon, L. J., Steffen, W., Jonsson, B. F., Folke, C., Falkenmark, M. and Johannessen, A.: Human modification of global water vapor flows from the land surface, Proc. Natl. Acad. Sci., 102(21), 7612–7617, doi:10.1073/pnas.0500208102, 2005.

INPE (Instituto Nacional de Pesquisas Espaciais): http://www.obt.inpe.br/prodes/index.php, last access: 19 July 2017.

Loarie, S. R., Lobell, D. B., Asner, G. P., Mu, Q., and Field, C. B.: Direct impacts on local climate of sugar-cane expansion in Brazil, Nature Clim. Change, 5 1, 105–109, 2011.

Miralles, D. G., Jiménez, C., Jung, M., Michel, D., Ershadi, A., McCabe, M. F., ... & Mu,

Q. (2016). The WACMOS-ET project-Part 2: Evaluation of global terrestrial evaporation data sets. Hydrology and Earth System Sciences, 20(2), 823-842.

Snyder, P. K.: The Influence of Tropical Deforestation on the Northern Hemisphere Climate by Atmospheric Teleconnections, Earth Interact., 14(4), 1–34, doi:10.1175/2010EI280.1, 2010.

Van der Ent, R. J., Tuinenburg, O. A., Knoche, H. R., Savenije, H. H. G., & Kunstmann, H. (2013). Should we use a simple or complex model for moisture recycling and atmospheric moisture tracking?. Hydrology and Earth System Sciences Discussions, 10 (5), 2013.

---

## Author Comment (AC1) · 11 Nov 2017

Response to comments by Referee#1 Dr. Patrick Keys on "Aerial and surface rivers: downwind impacts on water availability from land use changes in Amazonia"

We thank referee #1 Dr. Keys (hereafter the referee) for the suggestions and comments to help us improve the manuscript.

Weng et al. (hereafter, the authors) explore the role of land-use change on the hydrology of the Amazon, focusing on the implications of changes in evaporation on moisture recycling, precipitation, and subsequent runoff. The authors identify key regions in the Amazon that are particularly sensitive to changes in continental moisture recycling, and further identify how different land-use change scenarios can impact hydrology.

I find the study to be interesting, relevant, and timely. I recommend the paper receive minor revisions, prior to acceptance for publication in HESS. I will summarize my comments briefly, and then lay out more detailed comments below

The paper is on the cutting edge of land-use change & moisture recycling research, in the sense that it is examining sensitive regions to land-use change, the role of internal-watershed vs. external-watershed land-use change impacts, and the importance of different types of land-use change having very different types of consequences on hydrology. I encourage the authors to emphasize the 'cutting-edge' nature of their work a bit more, and be more bold in their conclusions.

We appreciate the positive feedback from the referee on the originality of the findings. We also perceive the underscoring of the novel part of our study helpful for communicating our results more efficiently to the community and have revised the conclusion (P13. L3-16) regarding this point.

I'm very interested in the development of the 'most influential precipitationshed' (MIP) concept. However, the MIP is essentially a definition of a precipitationshed boundary. The precipitationshed boundary that contains 100% of evaporation is the entire planet (at least in the context of the WAM-2layers). Thus, anything less than that domain requires the selection of a boundary. There is quite a bit of discussion on boundary selection and comparative advantages and disadvantages in the existing literature (e.g. the 70% boundary in Keys et al. (2012), the identification of the 'core precipitationshed' as a persistent, inter-annual source of moisture in Keys et al. (2014), and the discussion of the 1% boundary in Keys et al. (2017)). It is good that the authors are innovating on the concept of the precipitationshed, but I think the MIP should be put into better context as an approach for quantifying a boundary.

We agree with the referee that the precipitationshed boundary used in the manuscript needs further discussion in the context of previous studies. We added the citations suggested by the referee in Sect 2.1.3. "Previous studies have suggested and discussed different thresholds to delineate a precipitationshed boundary, e.g., 70% (Keys et al., 2012) or 1%

(Keys et al., 2017) threshold of continental recycled precipitation" We have also clarified the MIP definition in Sect. 2.1.3. The basic idea of the MIP is to emphasize the spatial heterogeneity within the precipitationshed. The 40% threshold that we choose to identify the MIPs in our study is due to the model resolution that we used (as the referee has also pointed out later) and the geographical region that we focused on. Our intention was also to approach a threshold that could be reasonable in land use experiments, since the larger threshold MIP of a given sink has larger total influence on the sink's rainfall, but has also a larger size, meaning that it is rather theoretically to have homogeneous land use change. Thus we have also improved the narrative for this part in Sect 2.1.3 by adding "In the present study, we propose a threshold that is a trade-off between the relative influence on the sink's rainfall and the size of the area where land use change could occur homogeneously.".

Regarding the MIP, I think that the authors need to emphasize more clearly that the 40% threshold is related to grid resolution (as far as I can tell, the only reference to this is at line 22-24).

Yes, it is possible that other studies operating on a finer grid resolution or focusing on different study regions can have a smaller threshold apply to all grid cells (P.5 L21-24). Our application of the 40% threshold in our study area appears plausible in reflecting important regions on moisture contribution to a given sink and can provide a hint for further studies operating on similar modelling resolutions and regions. We have included clarification on this both in Sect. 2.1.3 and Sect. 5.

Again, I think this paper is quite good, and is in need only of minor revisions before publication.

We thank the referee for the positive feedback on the manuscript and appreciate his suggestion that is helpful for improving the manuscript.

P2 L4 This sentence is confusing, especially the section ": : :operate under uncertainties of the undergoing land use change: : :". Please revise.

Revised as suggested.

P2 L24-25 I suggest the authors remove the part of the sentence ": : :,which has not been covered in depth by previous studies", since many studies have looked in detail at how land-use change might impact the hydrological regime in the Amazon. There is still much work to do of course, but there has still been quite a lot of research into land-use change, moisture recycling and the hydrological cycle throughout the Amazon.

We agree with the referee and removed the mentioned part in P2 L24-25.

P2 L22 Consider including Badger & Dirmeyer (2015) and Keys et al. (2016). Badger and Dirmeyer conducted a detailed examination of the climate impacts of land-use change in the Amazon, including a very detailed analysis of the hydroclimate. Keys et al. examine the role of vegetation change on moisture recycling, including a regional focus on a part of the

Amazon experiencing rapid land-use change (as well as using the WAM-2layers for the moisture tracking).

Thanks for the suggestion, we included the suggested literature and also added other references.

P2 L26 If this is the first instance of the abbreviation 'SDGs', please spell it out. Also, which of the 17 SDGs are the authors referring to? Consider adding some specificity here and a citation to support the relevance of moisture recycling (I'm not doubting its relevance, but it would be useful for the authors to chart this relevance more clearly and specifically).

Thank the referee for pointing this out, we revised it to make it more specific.

P5 L4 In section 2.1.3, the authors explain their concept of the 'most influential precipitationshed'.

From my understanding, this is simply a threshold-based boundary, correct? As stated earlier, this a very interesting idea, but the authors ought to acknowledge that this is one of several methods for delineating a precipitationshed boundary. I highlighted the previous studies in my General Comments that have discussed boundary methods. Essentially, the 40% MIP is the boundary which provides 40% of continentally recycled precipitation, correct?

The referee understood our analysis correctly. We agree with the referee's comment and have put it in the context of existing literature in both Sect. 2.1.3 and Sect. 4.4.

P6 L26 It would be useful to remind the reader that MIP essentially means the 40% terrestrial moisture recycling boundary (again, assuming I understand it correctly).

As the referee pointed out earlier, we have added clarification on the boundary context of the MIP threshold in Sect. 2.1.3. Thus we referred to Sect. 2.1.3 here and added the "40% threshold" description to avoid confusion. However, we decide to keep the MIP to hint to the readers the step's underlying purpose.

P7 L12 What is meant by "with high spatial efficiency"?

We mean stronger control per unit area and added this in the revision.

P7 L13 Why does the MIP account for 50% of the Amazonian evapotranspiration? Shouldn't it be 40%? Please clarify for easier interpretation of the result.

The Amazonian evapotranspiration contributes about 80% of the continental sourced rainfall in the sensitive areas. The MIP accounts for half of this 80% but has the size of 3.5% of the Amazon basin. We have added explanation in the revision.

- P8 L7 A bit confusing. Please replace "adding an extra time: : : on the original flow" with "more than doubling: : : the original flow"

- P8 L22 Perhaps replace "fashion" with "pattern"?

- Replaced as suggested.

P8 L30 Very interesting finding!

Thanks!

P9 L6-7 The finding about the rice planting not having very large impacts on run-off makes me curious about seasonal impacts (e.g. trees evaporate at different times than crops, etc.). Did you explore seasonal impacts? If so, please include some information on that analysis; if not, please include a few comments as to why it is outside the scope of this present work.

The seasonal variation was indeed outside the scope of our question since we focus on the spatial difference of land use change impacts on the annual water availability. We have stated in Sect.4.5 Limitation (P12L19) that future work focusing on specific purposes should take seasonal impacts into account.

P9 L20 "As it controls half the Amazonian evapotranspiration: : :" Again, I am confused about whether the MIP represents 50% or 40%.

The evapotranspiration from the Amazon basin contributes about 80% of the continentally sourced rainfall in the sensitive areas and we have revised it in P.7 to avoid confusion that would arise also in this part.

P10 L25 Do the authors mean "increases" where they wrote "increments"?

Yes, we changed that into "increases" in the revision.

P11 L12 The authors should consider citing Wang-Erlandsson et al. (2017) since they find these same types of results. Both the Wang-Erlandsson paper and this paper are currently in HESSD, and it would be useful as a reader to see they find complimentary results using a variation in methods. In the interest of full disclosure, I am a co-author on the Wang-Erlandsson et al. article, and will suggest to the lead author of that paper that they also cite this work (for the same reasons I suggested already).

We found the suggested discussion paper interesting as it finds similar effect from land use change on runoff through moisture recycling from global analysis. We added a citation to the manuscript.

P13 L2-5 Here the authors could be bolder in their conclusions about what is important and novel about their work. E.g. The importance of relatively small source areas for sensitive regions in the Amazon; also the importance of extra-basin land-use change on basin runoff.

We have revised Sect.5 Conclusion (P13. L3-16) to emphasize these points.

- P14 L1 Confusing sentence ": : :strong controls on the rainfall and runoff regimes of the sensitives." I think the authors are missing a word; perhaps "sensitive regions"?
- P14 L1-2 I recommend removing the final sentence since it is unnecessary.
- Fig 2 & 3 Both figures need a label on the colorbar
- Revised as suggested.

References

Keys, P. W., Van Der Ent, R. J., Gordon, L. J., Hoff, H., Nikoli, R. and Savenije, H. H. G.: Analyzing precipitationsheds to understand the vulnerability of rainfall dependent regions, Biogeosciences, 9(2), 733–746, doi:10.5194/bg-9-733-2012, 2012.

Keys, P. W., Wang-Erlandsson, L., Gordon, L. J., Galaz, V. and Ebbesson, J.: Approaching moisture recycling governance, Glob. Environ. Chang., 45, 15–23, doi:10.1016/j.gloenvcha.2017.04.007, 2017.

---

## Author Comment (AC2) · 11 Nov 2017

Response to comments by Referee#2 Dr. Rogier Westerhoff on "Aerial and Surface rivers: downwind impacts on water availability from land use changes in Amazonia"

We would like to thank referee #2 Dr. Westerhoff's (hereafter, the referee) suggestions that help us improve our manuscript.

This paper addresses the role of soil moisture recycling (aerial rivers) in the water balance, and how it affects downwind climates in scenarios of land-use, more specifically forest change in the Amazon. The paper reads well and touches on a topic that is not often discussed amongs hydrologist. The study also potentially has a large impact and therefore much relevance. I have to so that I am not an expert in soil moisture recycling and I could therefore not go into detail on validity of all the methods used. However, I do see that the paper could use some extra description and discussion on uncertainty.

We thank the referee for highlighting the potential application of our results to the community. We have included more discussion on uncertainty in the revised version as suggested to improve our narrative on the findings in Sect.4.5 limitation.

The only major comment I have is that the figures that you present incorporate some uncertainty or range, but without any explanation on how these were estimates. Also, other uncertainties are not addressed, e.g., the assumption of constant groundwater. Some uncertainties might be higher than your actual estimates. Or not, but without proper explanation we do not know. I think the paper also deserves a discussion that deals with uncertainty.

Since my comments are relatively easy to address (in my opinion) I therefore recommend this paper to be accepted with minor revisions.

Thank you to the referee for the positive feedback. We agree with the referee to improve the description and discussion on uncertainties in the revision to avoid potential confusion. We have added information to Table 2 where uncertainty ranges represent the standard errors of the measured data in Sakai et al., 2004. We also find it important to discuss uncertainties in the stable groundwater storage assumption and have added the discussion in Sect. 4.5. We have also added more information on the processes and methods (e.g. MIP boundaries) that we think needed more clarification.

These are my comments. Please treat the comments on uncertainty as less than minor.

-   Page 2, line 7. Two sentences that almost say the same, try turning these into one.
-   Page 2, line 19-21. The sentence is unclear, especially the part "some areas' water regime". Please rephrase.
-   Page 2, line 26. Explain the abbreviation SDGs. Maybe a reference to some of these SDGs (e.g. water)
-   Page 2, line 34. The first time you use the term aerial rivers, explain what they are. That is important, since the term is in the title. You can either probably solve that quite simple by saying:" 'aerial rivers', i.e., preferential pathway of moisture flow, termed in Arraut et al. (2012) as an analogy to surface rivers."

-   Revised as suggested.

Page 3, line 6-10. Try to avoid method description in the introduction.

We have now simplified this part in order to give the readers a brief impression of the tool used in the study, directly right after our introduction to the tools used in the community.

Page 3. Define what the sinks and sources are in this study.

We did not state it clearly and clarified our definition on P3 L15.

Page 5, line 22. It is e.g., not eg.

Corrected as suggested.

Page 6, line 13. I think you should explain the assumption of Zemp et al. (2014) in somewhat more detail, instead of the quick reference. What is balance? Are they equal? Or are their ratios equal?

We agree with the referee that the assumption was not clear and have clarified it. We further shifted the reference of Zemp et al. to Sect 4.1 to avoid confusion. We referred here to their paper originally for their discussion on the MOD experiments' E and P balance (Sect 2.1.2; Zemp et al., 2014). However, they did not make the assumption of E and P balance which was made here in the present study. Thus we shifted the reference to where we discuss the bias induced by the imbalance between E and P in some geographic regions e.g. the Andes (Sect. 4.1) as was also pointed out by Zemp et al. (2014) in their discussion.

- Page 6, line 13. The steady groundwater storage assumption is of major importance in my opinion. This needs to be in the discussion. For example, removing trees generally elevates the water table. Although the water table is already very shallow in most of the Amazon, small differences of e.g. 5 cm might have mahor differences in the whole balance that you are calculating. Can you something on the uncertainty surrounding that?
- Page 9, I would like to see some more uncertainty discussed. E.g. the groundwater assumptions. Also, it is not entirely clear from the method how you got your result uncertainty ranges (e.g., the 10-26%m 5-12% etc).
- We agree that it is important to discuss uncertainty in the assumption of stable groundwater storage in our runoff estimation. We have added a discussion (in Sect. 4.5) on studies partitioning river water storage and groundwater storage's contribution to the variation in terrestrial water storage. However, the groundwater storage's importance still remains disputed among previous studies focusing on the Amazon basin. We have referred to previous findings in the revision for the reader's information. In addition, we have included more relevant processes in our discussion in Sect. 4.5.

  The ranges in Sect. 3 Results were to express the number span between different land use change scenarios and we have added clarification (according to scenarios) when this expression first appeared to avoid confusion in the revision.

Page 8, line 6-7. a quarter of what, and extra time of what?. This sentence should be much clearer.

Clarified as suggested.

As also mention above, we have added more discussion on the uncertainties and limitation.

Page 9-10: Discussion: Can the discussion mention what the relative contribution is compared to the moisture from the sea?

The oceanic source is out of the scope of our study focusing on the terrestrial moisture recycling. However, we have added in the revision a description specifying that the terrestrial recycling contribution to the sensitive areas is 74.7% in average for readers' comparison.

Page 13, line 31. It is 'bottom-up'

Revised as suggested.

References

Sakai, R. K., Fitzjarrald, D. R., Moraes, O. L. L., Staebler, R. M., Acevedo, O. C., Czikowsky, M. J., Silva, R. Da, Brait, E. and Miranda, V.: Land-use change effects on local energy, water, and carbon balances in an Amazonian agricultural field, Glob. Chang. Biol., 10(5), 895–907, doi:10.1111/j.1529-8817.2003.00773.x, 2004.

Zemp, D. C., Schleussner, C. F., Barbosa, H. M. J., Van Der Ent, R. J., Donges, J. F., Heinke, J., Sampaio, G. and Rammig, A.: On the importance of cascading moisture recycling in South America, Atmos. Chem. Phys., 14(23), 13337–13359, doi:10.5194/acp-14-13337-2014, 2014.

---

## Author Comment (AC3) · 11 Nov 2017

Response to Mr. Coen Maas's comments
The paper discusses the impact of land use change on the hydrology of the Amazon. Land use change is happening a lot in the amazon these days and therefore, it is important to know the impact of this change on the regional and global scale. By using a moisture recycling tracking algorithm, Weng et al. tried to get a better understanding of the influence of land use change on rainfall, evapotranspiration and runoff. The results of the paper are that by all the land use changes the rainfall decreases. The extend of this change depends on the location where the land use changes happen. Furthermore, a change of the whole Amazon to a certain type of land use has a large influence on both the precipitation and the runoff. The paper touches a topic which is very relevant at the moment, other studies have been looking into this as well(Snyder, 2010, Gordon et al., 2005), but the spatial different sensitivity in the hydrological responses to land use change was not well understood. Like said in the paper itself, deforestation is happening in the Amazon to create agricultural land(INPE, 2017). This change to agricultural land use can have a massive impact on the hydrology in the Amazon. Due to the fact that the Amazon is such a large area, this could even have an influence on the world as well. Therefore, it is a very interesting topic which should be looked into even more with other researches. The paper is well written, but there are some minor improvements which could be made to make this better. These minor improvements will be stated later on in this review. Little research is done at this topic so the research which is conducted by Weng et al. is innovative. It is interesting because the outcomes of this research can be used in other areas which suffer from deforestation as well. For this reason this paper can be an eye opener for other people to investigate this process even more. The hydrological impact which is the result of this paper perfectly fits the aim of the Hydrology and earth system sciences (HESS) scientific journal. The research which is conducted is donewell, but there are some minor issues which could be solved. Therefore, I recommend some minor revisions before publications by HESS. The revisions that should be made in my opinion are listed below.

We appreciate the comments from Mr. Maas highlighting the original findings of our manuscript.

In my opinion there should be made a better substantiation of the use of MOD16ET data. The authors say: "Loarie et al. (2011) validated MOD16ET's estimation with eddy flux tower data and reported its good performance (differences in annual aver-age of evapotranspiration are less than 4 % in savannas, 5 % in tropical forests and 13 % in pasture agricultural lands)". However, other references say something else: "While all three evaporation products adequately represent the expected average geographical patterns and seasonality, there is a tendency in PM-MOD to underestimate the flux in the tropics and subtropics. Overall, results from GLEAM and PT-JPL appear more realistic when compared to surface water balances from 837 globally distributed catchments and to separate evaporation estimates from ERAInterim and the model tree ensemble (MTE)."(Miralles et al., 2016). These references are opposites of each other. The use of the MOD16ET method can have an uncertainty on all the figures and results that are made in this research.
I would suggest to make a better substantiation why the MOD16ET data is used and why for example the GLEAM or the PT-JPL were not used. Furthermore, a paragraph can be added to the discussion with the topic what the uncertainty of the MOD16ET is on the results that are made.
We thank Mr. Maas for the general comments discussing over the input data and model usage. We agree that better input data might exist but Miralles et al. (2016) also pointed out a generally good capture of geographical patterns and seasonality in ET among the three datasets. Since Miralles et al. have not specifically presented their results on the dataset's robustness at the Amazon basin for ET (though they have presented that for interception), we think it is still better to provide the validation on ET by Loarie et al. (2011) in the Amazon basin for reader's reference (P.4 L25-27). The interpretation and comparison between different input data is out of the scope of the presented research. However, we agree with Mr. Maas that uncertainties in the ET data might generate uncertainties in the recycling ratio and we have not specified that in the manuscript. It has already been discussed in Zemp et al. (2014) Sect 2.1.2., and we therefore referred the readers to such discussion in Sect. 4.1 (P.9L26-30) in the revision.

A second revision is to give more substantiation and discussion on the use of the WAM-2layers model. The WAM-2layers simulations of another experiment are used but the use of a WAM-2layers offline model give worse results than an online model like the RCM-tag model. In a paper by Van der Ent et al., 2013 a comparison is made between the WAM-2layer model and the RCM-tag model, a result of this comparison was that simulations of both models give globally the same result. However, at a regional scale, the error for the recycling ratio of the WAM-2layers model is relatively large if it is compared with this error of the RCM-tag model(respectively 2.8% against 1.9%(Van der Ent et al., 2013)). The research is mostly about the Amazon, which is a regional scale as well. Therefore, the results and figures could be different when a more precise method was used. I suggest the authors to take this into account in the discussion as well. The use of the WAM-2layers model has a larger uncertainty than an online model. Therefore, this uncertainty should be mentioned in the discussion.

We thank Mr. Maas who raises an important question related to the moisture tracking method. We decided to use the posteriori model because it can be based on observational data (as done in our study) (P.2L27) and it is less computational expensive compared to online models. Actually, we think that it wasn't concluded in van der Ent et al., 2013 which model was superior but they suggested avoiding usage of posteriori models at local scale. The recycling ratio provided by Mr. Maas was that in Lake Volta area and was not used in the indicated paper for interpreting regional study's results. Van der Ent et al., (2013) have suggested that the error was majorly from strong wind shear in West Africa thus we used the improved version (WAM-2layers) of the WAM model to decrease such errors in our estimation (P.4 L9-11). For the uncertainties from modelling choice, we have compared our results with the meta-analysis of the 44 GCM and RCM studies' results by Spracklen and Garcia-Carreras (2015) in P.10 L16-24. In our revised manuscript we have also added a reference to Table 2 in Zemp et al. (2014) that compares recycling ratios for the Amazon basin from the WAM-2layers model and other modelling approaches.

A third revision is the title of the manuscript is: "Aerial and surface rivers: downwind impacts on water availability from land use changes in Amazonia". This gives the feeling that the paper is about the water availability in the downwind areas. However, the conclusions that are stated in this manuscript are all about discharge and reduction of precipitation. If I look at the definition of water availability in a random dictionary I get the following: "The portion of a water resource that can be abstracted, as determined by the total water resource and the rights to abstract water from that water resource.".So the title will attract readers who are interested in the amount of water which is available in the amazon to abstract. The first sentence of the conclusion is: "From our analysis of the moisture recycling process, we conclude that Amazonian land use change's impacts on the water regime have spatial heterogeneity in two ways.". So the conclusion is about the water regime, not the availability. I would suggest that the title of the manuscript is going to be changed, especially the term "availability". This is a term which could attract the wrong readers. In my opinion, a term like "water regime" would fit better in this manuscript.

We thank Mr. Maas for this interesting suggestion and agree that there might be different interpretations on the title. However, according to FAO corporate document repository, http://www.fao.org/docrep/u5835e/u5835e03.htm, water availability is defined "The possibility of supplying as much water to the irrigation area... depends primarily on the availability of the water at its source...". We used the term because rainfall and runoff determines "the availability of the water at its source". We have considered Mr. Maas's suggestion carefully but decided to keep the title as it provides better hints to our discussions, especially Sect. 4.3 Water conservation hotspots out of watersheds and Sect. 4.4 Managing interconnected surface and aerial rivers crossing boundary. However, we have revised the conclusion (P13 L4-5 and P14L1) for better linkage of those ideas.

MINOR COMMENTS
- P2, line 7: place a space between 80 and the % sign
- P2, line 26: There is an abbreviation SDGs in this sentence but it is not said what this abbreviation means.

- P4, line 6-10: this is part of a methodology already. This should not be in the introduction
- P5, line 22: eg. Should be e.g.
- Fig 2: What are the units of the color bar? Give it a label.
- Fig 3: What are the units of the color bar? Give it a label.
- P11, line 19: "The results is: : :." Remove the "s" in the word "results".
- Revised as suggested.

P7, line 25 and 26: in the first line the reference to a figure is like: "Fig. 4" in the next sentence the reference is like: "figure. 5" Please be consistent. Use "Fig" or "Figure"
P8, line 15: "For that, we apply in the following: : :" remove "in".
Fig 1: Has no title in the figure itself.
Fig 4: Add a title at the figure itself
Fig 6: Try to give it the same mask as the other figures, now the whole of south America is showed while the results that are mentioned are only about the amazon.
- Thanks for the comments, we consider changes where appropriate.

References

van der Ent, R. J., Tuinenburg, O. A., Knoche, H.-R., Kunstmann, H., and Savenije, H. H. G.: Should we use a simple or complex model for moisture recycling and atmospheric moisture tracking?, Hydrol. Earth Syst. Sci., 17, 4869-4884, https://doi.org/10.5194/hess-17-4869-2013, 2013.

Loarie, S. R., Lobell, D. B., Asner, G. P., Mu, Q., and Field, C. B.: Direct impacts on local climate of sugar-cane expansion in Brazil, Nature Clim. Change, 1, 105–109, 2011.

Miralles, D. G., Jiménez, C., Jung, M., Michel, D., Ershadi, A., McCabe, M. F., Hirschi, M., Martens, B., Dolman, A. J., Fisher, J. B., Mu, Q., Seneviratne, S. I., Wood, E. F., and Fernández-Prieto, D.: The WACMOS-ET project – Part 2: Evaluation of global terrestrial evaporation data sets, Hydrol. Earth Syst. Sci., 20, 823-842, https://doi.org/10.5194/hess-20-823-2016, 2016.

Spracklen, D. V. and Garcia-Carreras, L.: The impact of Amazonian deforestation on Amazon basin rainfall, Geophys. Res. Lett., 42(21), 9546–9552, doi:10.1002/2015GL066063, 2015.

Zemp, D. C., Schleussner, C. F., Barbosa, H. M. J., Van Der Ent, R. J., Donges, J. F., Heinke, J., Sampaio, G. and Rammig, A.: On the importance of cascading moisture recycling in South America, Atmos. Chem. Phys., 14(23), 13337–13359, doi:10.5194/acp-14-13337-2014, 2014.